# Target-conditioned GFlowNet for Structure-based Drug Design

**Tony Shen**[*a], **Seonghwan Seo**[b], **Grayson Lee**[a], **Mohit Pandey**[e], **Jason Smith**[e], **Artem Cherkasov**[e], **Woo Youn Kim**[b,c,d], **Martin Ester**[*a]

[*] *Corresponding author. Contact: tony_shen_4@sfu.ca, ester@cs.sfu.ca*
[a] *School of Computing Science, Simon Fraser University, Burnaby, Canada.*
[b] *Department of Chemistry, KAIST, Daejeon, Republic of Korea.*
[c] *Graduate School of Data Science, KAIST, Daejeon, Republic of Korea.*
[d] *HITS Inc., Seoul, Republic of Korea.*
[e] *Vancouver Prostate Centre, University of British Columbia, Vancouver, Canada.*

**Reviewed on OpenReview:** *https://openreview.net/forum?id=N8cPv95zOU*

## Abstract

Searching the vast chemical space for drug-like molecules that bind with a protein pocket is a challenging task in drug discovery. Recently, structure-based generative models have been introduced which promise to be more efficient by learning to generate molecules for any given protein structure. However, since they learn the distribution of a limited protein-ligand complex dataset, structure-based methods do not yet outperform optimization-based methods that generate binding molecules for just one pocket. To overcome limitations on data while leveraging learning across protein targets, we choose to model the reward distribution conditioned on pocket structure, instead of the training data distribution. We design TacoGFN, a novel GFlowNet-based approach for structure-based drug design, which can generate molecules conditioned on any protein pocket structure with probabilities proportional to its affinity and property rewards. In the generative setting for CrossDocked2020 benchmark, TacoGFN attains a state-of-the-art success rate of 56.0% and $-8.44$ kcal/mol in median Vina Dock score while improving the generation time by multiple orders of magnitude. Fine-tuning TacoGFN further improves the median Vina Dock score to $-10.93$ kcal/mol and the success rate to 88.8%, outperforming all optimization-based methods.

## 1 Introductions

Structure-based drug design (SBDD) leverages target protein structures to search for high-affinity drug molecules. Due to the growing availability of protein structures from ML protein structure prediction methods (Jumper et al., 2021), and many novel targets identified from high-throughput perturbation experiments (Replogle et al., 2022), SBDD is becoming an increasingly powerful approach in drug discovery. It currently takes 13-15 years and between US $2 billion and $3 billion for a single drug to be developed and approved (Pushpakom et al., 2018). The substantial expense and time of drug development not only impose a significant burden on healthcare systems but also amplify societal risks during global health crises, such as COVID-19. There is an urgent need to expedite the design of novel drug candidates for new protein targets.

Traditionally, virtual screening has been used to discover binding molecules by predicting supramolecular interactions between ligands and a target protein with molecular docking. Its efficacy is impeded by the exhaustive nature of its search, and by the high computational cost of molecular docking. To overcome this challenge, many optimization-based methods (Bengio et al., 2021; Fu et al., 2022; Lee et al., 2023; Reidenbach, 2024) have been proposed to generate high-affinity molecules for one protein pocket only. These methods typically do not take protein structure as input and use docking measures as the reward. Recently, structure-based generative models (Luo et al., 2021; Peng et al., 2022; Guan et al., 2023b) have been proposed

to design molecules (ligands) conditioned on the pocket structure. These methods learn the distribution of a training dataset of protein-ligand complexes. Structure-based generative model has the advantage of learning generalized protein-ligand interaction patterns by leveraging different pocket structures during training.

However, due to the high cost of the experiments, the size of the training datasets for SBDD, i.e. high-quality protein-ligand binding structure data, is relatively small. PDBBind (Liu et al., 2014), the underlying dataset of the standard CrossDocked2020 benchmark (Francoeur et al., 2020), contains only 19,443 protein-ligand complexes. After removing common biomolecules (lipids, peptides, carbohydrates, and nucleotides) and duplicates, only 4,200 unique drug-like molecules remain (Powers et al., 2023). This is only a tiny fraction of the entire chemical space of drug-like molecules that is assumed to consist of $\sim 10^{60}$ molecules (Lipinski et al., 1997). As a consequence, existing structure-based generative models relying on data-distribution learning model only a very small part of the overall chemical space and have struggled to generate novel molecules with significantly improved properties (Lee et al., 2023); Therefore, existing structure-based generative methods do not yet outperform optimization-based methods on a single target.

In summary, many current optimization-based methods cannot leverage learning from across protein pocket structures - they are limited to a reward function based on affinity to one single protein target per model. On the other hand, while structure-based models can learn generalized protein-ligand interaction patterns by training on different pockets, they are restricted to modelling a limited data distribution. Therefore they are unable to effectively explore the wider chemical space for desirable molecules, and under perform optimization-based methods. Our goal in this paper is to both leverage learning across protein target structures, and overcome the challenge of limited data distribution. To the best of our knowledge, TacoGFN is the first RL model to address the challenging task of modelling a family of reward functions induced from all pocket structures.

In this paper, we frame the task of structure-based molecule generation as learning a reward distribution instead of a training data distribution, and adopt GFlowNet Bengio et al. (2021) - an energy-based generative model for generating combinatorial objects. We propose TacoGFN, a _Target Conditioned Generative Flow Network_ that generates molecules conditioned on any given protein pocket structures, guided by affinity to the pocket, drug-likeness and synthesizability measures. This formulation allows us to explore the greater chemical space as we are no longer constrained to a fixed dataset. We performed an experimental evaluation on the standard CrossDocked benchmark dataset and demonstrated that TacoGFN clearly outperforms state-of-the-art structure-based generative methods, improving the success rate to 56.0% from the previous best of 24.5%. TacoGFN shows even stronger performance with fine-tuning (TacoGFN+FT), resulting in -10.93 kcal/mol in median Vina Dock score and up to 88.8% in success rate, outperforming all optimization-based models.

To summarize, the main contributions of this paper are:

- We point out the limitations of previous data-based distribution learning, and instead propose to frame structure-based molecule generation as learning from the reward distribution. To this end, we propose TacoGFN, the first application of GFlowNet for learning a family of molecule distributions conditioned on protein pocket structures.

- We introduced a novel pharmacophore-based affinity predictor, with improved generalization capabilities using a pre-trained pharmacophore representation, enabling fast affinity evaluation.

- TacoGFN clearly outperforms the state-of-the-art structure-based methods on the Cross-Docked2020 benchmark and demonstrates the benefit of reward-based distribution learning.

- TacoGFN+FT outperforms the optimization methods focused on a single pocket only, learning from diverse molecules for various protein pocket structures. TacoGFN+FT attains the best Vina Dock score, high-affinity rate and success rate among all optimization-based methods.

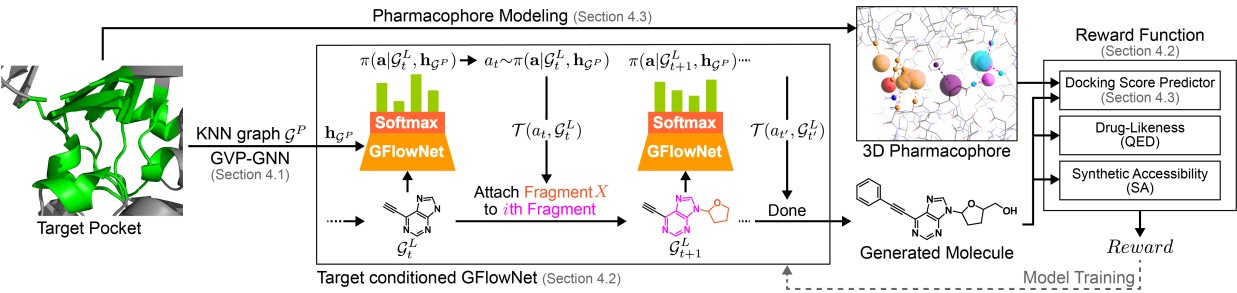

Figure 1: Overview of the sampling and training process of TacoGFN.

# 2 Related Work

**Structure-based molecular generation** aims to generate high-affinity molecules for any target pocket. They are expected to generalize on previously unseen test pockets, and therefore, do not have access to the docking oracle for the test pockets during inference or training time. The goal for evaluating the quality of molecules generated for unseen protein structures is to measure whether SBDD models have learned generalized protein-ligand interaction patterns during training. Many approaches for this problem setting has been proposed. LiGAN (Masuda et al., 2020) uses 3D CNNs to encode the protein pocket structure and predict atom densities from the encoded latent space. 3DSBDD (Luo et al., 2021) and Pocket2Mol (Peng et al., 2022) adopt an auto-regressive approach to generate molecules atom by atom. Other methods such as FLAG (Zhang et al., 2023b) and DrugGPS (Zhang & Liu, 2023) build molecules fragment by fragment to leverage the chemical prior. A very recent line of research employs diffusion models (Guan et al., 2023a;b; Schneuing et al., 2023) for SBDD. TargetDiff (Guan et al., 2023a) is a diffusion-based method which generates atom coordinates and atom types in a non-autoregressive way, and bonds are generated in a post-processing step. DecompDiff (Guan et al., 2023b) is a diffusion model which generates both atoms and bonds with decomposed priors, which reflect the natural decomposition of a ligand molecule into arms and scaffold. A recent analysis paper (Harris et al., 2023) questions the assumption that explicit 3D modelling of the ligand improves performance, after finding a much higher occurrence of physical violations and fewer key interactions in molecules generated using 3D modelling. In our work, we generate molecules in 2D space to vastly reduce the search space and compute time; Our approach is shown to effectively leverage the target pocket structure for generating molecules with high affinity.

**Optimization-based molecular generation** aims to generate molecules that satisfy certain optimization goals. Compared to the distribution-based generative model, these methods can be designed to optimize for molecules with strong binding affinity to a particular protein target. Since they have access to a docking oracle during training or inference time, they are not directly comparable to the structure-based generative problem setting. Reinforcement Learning (RL) methods such as ReLeaSE (Olivecrona et al., 2017), MolDQN (Zhou et al., 2019) and REINVENT (Blaschke et al., 2020) have been proposed to guide the generation of molecules toward desirable properties. MORLD (Jeon & Kim, 2020) and MoleGuLAR (Goel et al., 2021b) combine RL and docking calculations to design novel ligands. RGA (Fu et al., 2022) proposes a variant of a genetic algorithm that is guided by reinforcement learning and is pre-trained on multiple protein target structures. MOOD (Lee et al., 2023) incorporates out-of-distribution and property-guided exploration in diffusion models for goal-directed molecule generation. DecompOpt Zhou et al. (2024) combines a pre-trained structure-based equivariant diffusion model with a docking-based greedy iterative optimization loop. EvoSBDD Reidenbach (2024) improves efficiency by performing black-box optimization over the 1D latent space using a docking oracle. In summary, optimization-based molecular generation evaluates the performance of the algorithm on optimizing for seen targets. Compared to the structure-based setting, which requires modelling the molecule distribution conditional to any protein pocket structure and generalizing to unseen protein pockets, optimizing the molecule distribution for a single protein pocket is a less challenging task. We refer readers to a more detailed comparison of the problem setting in Appendix B.

**Protein-ligand affinity prediction.** Predicting the affinity or the docking score of a ligand to a target, in the absence of their binding complex structure, is a difficult task. Most previous docking score prediction models have been limited to a single protein target (Bengio et al., 2021; Gentile et al., 2020). Therefore, they are not suitable for the aim of designing high-affinity molecules for any given protein pocket structure. Recently, several methods have been proposed to predict ligand affinity for arbitrary protein targets (Zhang et al., 2023a; Pandey et al., 2022). However, these approaches are prone to memorizing the structural bias instead of learning the physics of protein-ligand binding and show low generalization ability to unseen ligands or proteins (Wallach & Heifets, 2018; Chan et al., 2023). To this end, we adopt a pre-trained pharmacophore representation, which only models the key interaction sites for a protein pocket. This prior improves our docking score predictor's ability to learn physical interactions and generalize to unseen data.

## 3   GFlowNet Preliminaries

Generative Flow Networks (GFlowNets, GFN) (Bengio et al., 2021) learn a stochastic policy $\pi$ for generating a combinatorial object (such as molecular graph) $x \in \mathcal{X}$. The probability of constructing $x$, denoted as $\pi(x)$, is trained to be proportional to a non-negative reward function $R : \mathcal{X} \mapsto \mathbb{R}^+$ defined on the space $\mathcal{X}$. This property of GFlowNet is ideal for generating diverse molecules with desirable properties. Conditional GFlowNet introduced in (Jain et al., 2023) simultaneously models a family of reward functions. Each conditional information, denoted as $c \in C$, induces a unique reward function $R(x|c)$. In our work, we adopt conditional GFlowNet for SBDD settings, by encoding target pocket structure as condition $c$, where $c$ is a high dimensional representation of the pocket structure. Thus, a single GFlowNet models high-reward molecule distribution across all protein pockets.

Each object $x$ is constructed from a sequence of actions $a \in \mathcal{A}$. In molecular settings, a molecule is constructed by inserting molecule fragments into a partially constructed fragment graph state $s \in \mathcal{S}$ (Bengio et al., 2021). Conceptually, a GFlowNet is an acyclic graph $G = (\mathcal{S}, \mathcal{E})$, with nodes $\mathcal{S}$ and edges $\mathcal{E}$. Each transition $s \to s' \in \mathcal{E}$ via action $a \in \mathcal{A}$ corresponds to an edge in graph $G$. The transition function $\mathcal{T} : \mathcal{S}, \mathcal{A} \mapsto \mathcal{S}$ computes the new state $s' = \mathcal{T}(s, a)$ given action $a$ on state $s$. A special action moves state $s$ into terminating states $\mathcal{X} \subset \mathcal{S}$. We define the initial empty graph as $s_0$. Construction of $x$ can be defined over a trajectory of states $\tau = (s_0 \to s_1 \to \ldots \to x)$.

Following previous studies, we introduce an exponent $\beta$ to the reward function $R(x|c)$, thus modelling $\pi(x|c, \beta) \propto R(x|c)^\beta$. This steers generating probability distribution to focus on the modes of $R(x|c)$, which is crucial for producing candidates that are high in reward. Adjusting $\beta$ allows us to manage the balance between diversity and achieving higher rewards.

The forward transition probability $P_F(s'|s, c, \beta)$ of a GFlowNet represents the probability distribution of reaching state $s'$ from state $s$ conditioned on context $c$ and reward temperature $\beta$. Partition function $Z(c, \beta)$ is the sum of the rewards $R(x|c)^\beta$ for all objects $x \in \mathcal{X}$ under context $c$. We adopt the Trajectory Balance objective from Equation 1 to efficiently learn a forward transition policy $P_F$ that generates object $x$ with probability proportional to its reward $R(x|c)^\beta$ (Malkin et al., 2023).

$$\mathcal{L}_{TB}(\tau, c, \beta; \theta) = \left( \log \frac{Z_\theta(c, \beta) \prod_{s \to s' \in \tau} P_{F_\theta}(s'|s, c, \beta)}{R(x|c)^\beta} \right)^2 , \tag{1}$$

$\theta$ represents the learnable parameters. Our goal is to model the probability $\pi(x|c, \beta) \approx \frac{R(x|c)^\beta}{Z(c,\beta)}$ across all molecules $x \in \mathcal{X}$, protein pocket contexts $c \in \mathcal{C}$ and various reward temperatures $\beta$.

## 4   TacoGFN

TacoGFN is a structure-based molecular generative model that generalizes over all protein pockets with a single model. By matching the reward distribution instead of limited data distribution, our method explores the greater chemical space to generate high-affinity molecules with properties desirable as a drug candidate. Furthermore, by encoding pocket structure information, TacoGFN and its fine-tuned variant are able to leverage learning from diverse protein pocket structures.

**Problem definition.** The goal of structure-based drug design (SBDD) problem is to generate molecules with both desirable properties and strong binding affinity with respect to any given protein pocket structure. The goal for SBDD models is learning generalized protein-ligand interaction patterns during training. Therefore, SBDD models take pocket structures as input and are expected to generalize for unseen pocket structures. We refer readers to a detailed contrast of the SBDD problem setting with the optimization setting in Appendix B

The pocket structure is denoted as $P$ and will be represented as a $K$-nearest neighbour (KNN) residue graph. The ligand is denoted as $L$ and will be represented either as an atom graph or a fragment graph. We define a reward function $R(L|P)$ based on a molecule's predicted docking score, drug-likeliness and synthesizability. Our goal is to learn a molecule generation policy that constructs molecules $L$ given protein structure $P$ with probability matching their reward (exponentiated with $\beta$), such that $\pi(L|P, \beta) \propto R(L|P)^\beta$.

**Method overview.** The protein pocket is first featurized as a $K$-nearest neighbor residue graph and encoded using GVP-GNN (Jing et al., 2021) (section: 4.1). Then, we use this pocket embedding as the condition for GFlowNet molecule generation (section 4.2). Finally, we reward generated molecules using our docking score prediction model which leverages protein-ligand interaction priors (section 4.3). The high-level architecture of our SBDD framework is illustrated in Figure 1.

## 4.1 Pocket structure encoder

We represent structure of the pocket $P$ as a standard $K$-nearest neighbor (KNN) residue graph $\mathcal{G}^{\mathcal{P}} = (\mathcal{V}^{\mathcal{P}}, \mathcal{E}^{\mathcal{P}})$ - following previous work in protein representation (Jing et al., 2021). The $i$-th residue node $v_i^{\mathcal{P}} \in \mathcal{V}^{\mathcal{P}}$ is featurized using its geometric and chemical properties. These features include the type of residue, the dihedral angles of the atoms in the residue backbone, and the directional unit vectors. An edge $e_{ij}^{\mathcal{P}} \in \mathcal{E}^{\mathcal{P}}$ is formed if the $j$-th residue $v_j^{\mathcal{P}}$ is among the $K$-nearest neighbors of residue $v_i^{\mathcal{P}}$, as measured by the euclidean distance between their respective $C_a$ atoms. We set the number of neighbours $K = 30$. An edge is featurized with the Euclidean distance, distance along the backbone and the direction vector between the two residues. These features sufficiently describe the features of the protein pocket.

We apply a graph neural network with geometric vector perceptrons (GVP) layers (Jing et al., 2021) to the KNN pocket graph $\mathcal{G}^{\mathcal{P}}$ to learn the node embedding $\mathbf{h}_{v_i^{\mathcal{P}}}$ for each residue. The node embeddings $\{\mathbf{h}_{v_i^{\mathcal{P}}}\}$ are then averaged to obtain an embedding of the entire graph $\mathbf{h}_{\mathcal{G}^{\mathcal{P}}}$. We use GVP because it encodes the protein pocket into an embedding that is invariant to rotations and translations.

## 4.2 Pocket conditioned GFlowNet

In this section, we discuss how to employ the pocket structure, more specifically its latent embedding, to condition the GFlowNet to generate molecules that interact with a given protein pocket. Furthermore, we describe our fragment-based molecular generation framework.

**Molecule representation.** During molecular generation, we represent ligands as a 2D molecular graph $\mathcal{G}^L = (\mathcal{V}^L, \mathcal{E}^L)$ with node $v_i^L \in \mathcal{V}^L$ representing a molecule fragment, and directional edges $e_{ij}^L \in \mathcal{E}^L$ indicating the attachment atom of fragment $v_i^L$ that connects to fragment $v_j^L$. Since a molecule's reward (representing its desirability as a real-world drug) should be the same regardless of its predicted 3D conformation, we represent ligands as 2D graphs here. [1]

**Fragment vocabulary construction.** TACOGFN generates molecules by adding one molecular fragment at a time. To create the vocabulary of fragments used, we extract common fragments from a chemical database in a data-driven and chemically valid way. To obtain a fragment vocabulary, we first apply BRICS decomposition (Degen et al., 2008) to 250k ZINC20 (Irwin et al., 2020) molecules. BRICS breaks molecules

---

[1]Docking score is dependent on a molecule's 3D conformation. However, we consider the best docking score for a molecule as the reward here, which is invariant to its conformation in this context. Taking the best docking score is a valid approach, because when a compound is tested for binding in the wet lab, the compound simply binds the protein with the conformations which results in the strongest affinity. Furthermore, the predicted 3D position of generated molecules from existing diffusion-based SBDD models often changes significantly upon re-docking and likely do not reflect the true conformation. (Reidenbach, 2024).

via retrosynthetic rules and provides synthetically accessible building blocks, i.e. molecules that are easy to prepare (Seo et al., 2023). To reduce the fragment vocabulary size, we further break all single bonds connecting a heavy atom to a ring structure. Next, we retain the fragments that occur in more than 50 (or 0.02%) of the molecules in our ZINC set. Only atoms (of a fragment) in the bonds decomposed by the BRICS can join to form new fragment connections during generation time; This reduces the occurrence of non-synthesizable attachment points forming bonds. This results in a set of 475 building block-like fragments. As our fragments are mined from a synthetically accessible virtual library and poor synthetic accessibility is penalized in the reward function, TACOGFN achieves superior synthetic accessibility compared to all existing SBDD methods.

**Molecular generation framework.** We formulate molecular generation as a sequential decision process and implement it using a GFlowNet. At the $t$-th step, the forward action policy $P_a$ samples an action $a$ depending on the current molecule state $\mathcal{G}_t^L$ and pocket embedding $\mathbf{h}_{\mathcal{G}^{\mathcal{P}}}$. The transition function $\mathcal{T}$ is a deterministic function which applies the action to the molecular graph at step $t$ to produce a molecule graph at step $t+1$.

$$a_t \sim P_a(\mathbf{a}|\mathcal{G}_t^L, \mathbf{h}_{\mathcal{G}^{\mathcal{P}}}) \tag{2}$$

$$\mathcal{G}_{t+1}^L = \mathcal{T}(a_t, \mathcal{G}_t^L) \tag{3}$$

Previous autoregressive models often formulate action prediction as a supervised task, where the goal is to predict the correct ground truth actions obtained from masked molecules (Peng et al., 2022). Instead, our molecule generation policy $P_a$ aims to generate molecules with probability $P(\mathcal{G}^L|\mathbf{h}_{\mathcal{G}^{\mathcal{P}}})$ proportional to the reward.

**Pocket conditioned molecular generation.** Here, we adopt the architecture for molecular generation introduced in Multi-Objective GFlowNet (MO-GFN) (Jain et al., 2023). Instead of conditioning the GFlowNet on multi-objective preference, we condition the GFlowNet on pocket embeddings to learn a family of molecular distributions corresponding to a family of reward functions induced from the pocket structure diversity.

We use a graph transformer (Yun et al., 2020) to model the probability $P_a(\mathbf{a}|\mathcal{G}_t^L, \mathbf{h}_{\mathcal{G}^{\mathcal{P}}})$, by taking the partially constructed molecular graph at the $t$-th time step $\mathcal{G}_t^L$ and the pocket embedding $\mathbf{h}_{\mathcal{G}^{\mathcal{P}}}$ as input. The input feature $\mathbf{h}_i^{L(0)}$ of molecular node $v_i^L$ is a one-hot encoding of the node's fragment type. The molecular edge input feature $\mathbf{e}_{ij}^{L(0)}$ is a one-hot mapping of the attachment atom index in node $v_i^L$ which connects to node $v_j^L$. We add an additional virtual conditioning node $\mathbf{h}^{V(0)}$, featurized using pocket embedding $\mathbf{h}_{\mathcal{G}^{\mathcal{P}}}$, to the graph $\mathcal{G}_t^L$ (Pham et al., 2017). This virtual node is connected to all other nodes and serves as a graph-level node to provide pocket information. After $N$ Transformer layers, the set of final node embeddings $\{\mathbf{h}_i^{L(N)}\}$ and edge embeddings $\{\mathbf{e}_{ij}^{L(N)}\}$ are obtained. The final graph level embedding $\mathbf{g}^{L(N)}$ is obtained via the concatenation of the global average pooling of the node embeddings and the final virtual node embedding $\mathbf{h}^{V(N)}$, as seen in Equation 4.

$$\mathbf{g}^{L(N)} = Concat\left(AvgPool\left(\left\{\mathbf{h}_i^{L(N)}\right\}\right), \mathbf{h}^{V(N)}\right) \tag{4}$$

Using these final molecular graph embeddings, we follow the actions defined in previous works on fragment-based molecular generation (Bengio et al., 2021; Jin et al., 2018). Full details can be found in Appendix C.1.1.

**Reward function.** A drug candidate must not only have a high affinity to the pocket but also satisfy drug-like properties and synthetizability requirements to be selected for experimental validation. We design a reward function by multiplying the normalized score from all three aspects, using QED (Bickerton et al., 2012) as a measure of drug-likeliness, SA (Ertl & Schuffenhauer, 2009) as a measure of ease-of-synthesizability, and predicted docking score as a measure of affinity between ligand and protein[2]. In drug discovery, it is crucial to design molecules that simultaneously satisfy all of these relevant properties, despite the inherent trade-offs among these properties. By multiplying the scores, the reward function ensures that a low score in

---

[2]Computing the reward is very fast, for example, computing it for a batch of 64 protein-ligand pairs took under 0.15 seconds.

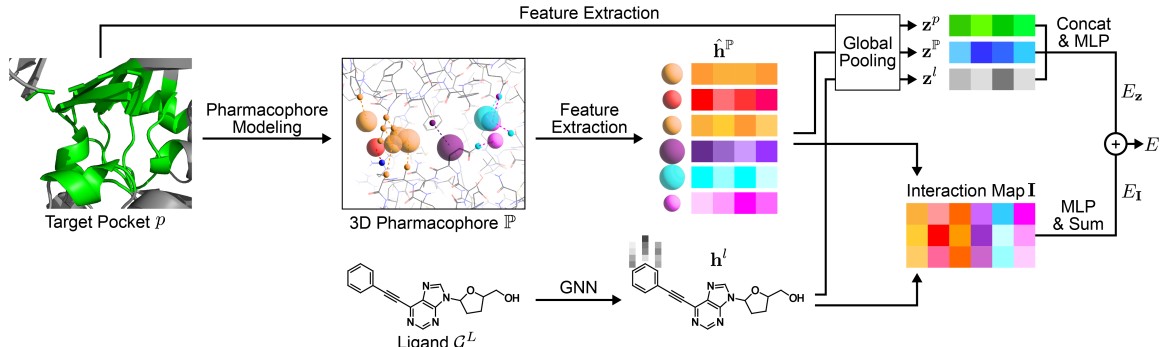

Figure 2: Model architecture of the docking score predictor. Each pharmacophore point is represented as a sphere and corresponds to a desired ligand characteristic for a binding interaction.

any one of the factors (QED, SA, or DS) will significantly reduce the overall reward. Therefore, multiplying the scores is a more appropriate choice than summing the scores here.

We just have to optimize QED and SA up to a certain threshold for a molecule to be suitable as a drug candidate - optimizing them further does not bring additional utility (Coley, 2021). Therefore, we clip the reward for the QED or SA component to 1 when they achieve their respective threshold $t_{QED}$ and $t_{SA}$. For example, the reward function will prioritize more on optimizing QED if $t_{QED}$ is higher, at the expense of other properties such as affinity (See details of reward function in Appendix C.1.2).

### 4.3 Docking score predictor with pharmacophore prior

The exploration of chemical space with a GFlowNet requires evaluating binding affinities for millions of molecules sampled during training. However, using molecular docking for evaluation is computationally expensive. Here, we propose a fast ML-based docking score predictor that generalizes well across molecule and protein pocket distributions, as described in Figure 2 and Equation 5. (See details in Appendix C.2)

We leverage PharmacoNet (Seo & Kim, 2023), a recent deep learning method to obtain the pharmacophore $\mathbb{P}$[3] from a pocket structure $P$. To enrich the representation, we obtained the embedding of the pocket $\mathbf{z}^P$ and the embeddings of pharmacophore points $\{\hat{\mathbf{h}}_i^{\mathbb{P}}\}$ from PharmacoNet. The 1D representation vectors of the pharmacophore $\mathbf{z}^{\mathbb{P}}$ is computed by the global pooling of $\{\hat{\mathbf{h}}_i^{\mathbb{P}}\}$ .

During docking score prediction, we represent a ligand as an atom-level 2D molecular graph. We apply Graph Isomorphism Network (Hu et al., 2019) to the atom graph to obtain the node embeddings $\{\mathbf{h}_j^L\}$. $\mathbf{z}^L$ is a 1D representation vector of the ligand, obtained by the global pooling of node embeddings $\{\mathbf{h}_j^L\}$. Then, we incorporate a pairwise interaction map $\mathbf{I}$ - computed from the outer product of $\{\hat{\mathbf{h}}_i^{\mathbb{P}}\}$ and $\{\mathbf{h}_j^L\}$. This preserves the structural topology and binding interaction details. Notably, our docking score predictor uses the pharmacophore points involved in binding instead of atoms or amino acids to obtain the interaction map. It improves the generalization ability at a reduced computational cost through coarse-grained modelling of the pocket at the pharmacophore level.

$$E = \phi_z(Concat(\mathbf{z}^P, \mathbf{z}^{\mathbb{P}}, \mathbf{z}^L)) + SumPool(\phi_{\mathbf{I}}(\mathbf{I})) \tag{5}$$

---

[3]Pharmacophore is a point set, where each point describes the desirable motif properties (such as being aromatic) a ligand should possess at this geometric position to form energetically or entropically favourable interactions (e.g. $\pi$–$\pi$ stacking) with the protein target. (Wermuth et al., 1998; Yang, 2010).

# 5 Experiments

## 5.1 Dataset

We train and evaluate TacoGFN on the commonly used **CrossDocked** benchmark (Francoeur et al., 2020). We first apply the splitting and processing protocol on the CrossDocked dataset to obtain the same train and test split of the 100k protein-ligand pairs as previous methods (Luo et al., 2021; Peng et al., 2022; Guan et al., 2023b). (See details in Appendix D.1). We then train our docking score predictor on the training split of the protein-ligand pairs to predict their corresponding Vina Dock scores. Since TacoGFN is trained on the same set of protein-ligand pairs and evaluated on the same unseen pockets, our experimental results can be directly compared against the published results of existing structure-based generative models.

## 5.2 Training

We train one TacoGFN model to generate molecules conditional to any protein target structures using predicted affinity, Synthetic Accessibility (SA), and drug-likeness (QED) as the reward. We also adopt Double GFN (Lau et al., 2023) to improve exploration in sparse reward domains and high-dimensional states, by initializing two networks to model the action policy: an online network and a target network.

For each training trajectory, one protein pocket is randomly drawn from the CrossDocked training set first. Then, a molecule is generated through sampling a sequence of actions using our target network which models the forward action policy $P_{target}$. The reward is then computed for that molecule with respect to the protein target. The Trajectory Balance loss (Equation 1) is used to train the online network modelling the action policy $P_{online}$. This loss has the objective for our model to generate objects with probabilities proportional to the rewards (consisting of predicted docking score, QED and SA). Periodically, the weights of the target network are updated by the weights of the online network using a delayed strategy. This strategy reduces training instability and promotes explorations of the larger chemical space.

## 5.3 Evaluation

In all evaluations, each structure-based generative model is tasked to produce 100 molecules (ligands) for each of the 100 unseen protein pockets from the CrossDock-100k test set.

**Evaluation metrics.** We adopt the following commonly used metrics from Guan et al. (2023a) and Reidenbach (2024): (1) **Validity** is the percentage of unique generated molecules free of reconstruction errors and disconnections as determined by RDKit. (2) **Vina Dock** approximates the binding energy between a generated molecule and a protein pocket, where a lower docking score indicates a higher binding affinity. (3) **High Affinity** measures the percentage of generated molecules with higher affinity than the reference molecule. (4) **QED** is a measure of drug-likeness, estimating a molecule's suitability as an oral drug based on its properties (Bickerton et al., 2012). (5) **Synthetic Accessibility (SA)** estimates difficulty of synthesizing the molecule (Ertl & Schuffenhauer, 2009). The score is normalized between 0 and 1 using the formula $(10 - SA)/9$. (6) **Diversity** is calculated as the average pairwise fingerprint Tanimoto distance between molecules generated for a pocket. (7) **Success Rate** is the percentage of molecules which pass the same criteria (QED > 0.25, SA > 0.59, Vina Dock < -8.18) as in Long et al. (2022); Guan et al. (2023b); Zhou et al. (2024); Reidenbach (2024). (8) **Time** is the average runtime (in seconds) for generating 100 unique and valid molecules for a pocket.

**Baselines and problem settings.** Similar to Zhou et al. (2024) and (Reidenbach, 2024), we separate existing methods by their problem definitions:

1) **Generative** methods are expected to generalize for pocket structures unseen during training. They generate molecules for test pockets at inference time without optimization loops or access to docking programs. Therefore, methods under this setting are only allowed to generate 100 molecules for each pocket in one-shot. We compare **TacoGFN** trained on CrossDocked-100k against the following generative models: **liGAN** (Ragoza et al., 2022), **GraphBP** (Liu et al., 2022), **AR** (Luo et al., 2021), **Pocket2Mol** (Peng et al., 2022), **TargetDiff** (Guan et al., 2023a), **DiffSBDD** (Schneuing et al., 2023), **DecompDiff** (Guan

Table 1: Comparison of the properties of molecules generated in the **generative** problem setting for the CrossDocked test set pockets. The reference molecules are from the CrossDocked test set. The best results are in **bold**. The average and median values are calculated over the averages for each pocket. The prior method results are taken from their publication.

| Model | Validity (↑) Avg. | Vina Dock (↓) Avg. | Med. | High Affinity (↑) Avg. | Med. | QED (↑) Avg. | Med. | SA (↑) Avg. | Med. | Diversity (↑) Avg. | Med. | Success Rate (↑) Avg. | Time (↓) Gen |
|---|---|---|---|---|---|---|---|---|---|---|---|---|---|
| Reference | - | -7.45 | -7.26 | - | - | 0.48 | 0.47 | 0.73 | 0.74 | - | - | 25.0% | - |
| liGAN | - | -6.33 | -6.20 | 21.1% | 11.1% | 0.39 | 0.39 | 0.59 | 0.57 | 0.66 | 0.67 | 3.9% | - |
| GraphBP | - | -4.80 | -4.70 | 14.2% | 6.7% | 0.43 | 0.45 | 0.49 | 0.48 | **0.79** | **0.78** | 0.1% | 10 |
| AR | 92.95% | -6.75 | -6.62 | 37.9% | 31.0% | 0.51 | 0.50 | 0.63 | 0.63 | 0.70 | 0.70 | 7.1% | 19659 |
| Pocket2Mol | 98.31% | -7.15 | -6.79 | 48.4% | 51.0% | 0.56 | 0.57 | 0.74 | 0.75 | 0.69 | 0.71 | 24.4% | 2504 |
| TargetDiff | 90.35% | -7.80 | -7.91 | 58.1% | 59.1% | 0.48 | 0.48 | 0.58 | 0.58 | 0.72 | 0.71 | 10.5% | 3428 |
| DiffSBDD | 85.01% | -8.03 | -7.75 | 55.3% | 56.6% | 0.47 | 0.47 | 0.55 | 0.56 | 0.76 | 0.76 | 6.0% | 160 |
| DecompDiff | 71.96% | **-8.39** | -8.43 | 64.4% | 71.0% | 0.45 | 0.43 | 0.61 | 0.60 | 0.68 | 0.68 | 24.5% | 6189 |
| TacoGFN (Ours) | **100%** | -8.24 | **-8.44** | **67.5%** | **92.0%** | **0.67** | **0.67** | **0.79** | **0.79** | 0.53 | 0.53 | **56.0%** | 4 |

et al., 2023b). For consistency with existing baselines, molecules are docked using QVina in this problem setting (Trott & Olson, 2010; Alhossary et al., 2015). For details on the docking protocol for the generative setting, please see Appendix D.2.

2) **Optimization** methods are able to leverage docking on the target pocket. Unlike the generative setting, these methods typically iteratively optimize the candidate pool and select the top 100 molecules. We compare against the following methods: **RGA** (Fu et al., 2022) and **EvoSBDD** (Reidenbach, 2024) use a black-box algorithms to conduct rounds of the optimization process, with molecule fitness based on docking score to the target pocket. **DecompOpt** and **TargetDiff+Opt** (Zhou et al., 2024) optimizes molecules generated by pre-trained SBDD models via rounds of optimization process involving re-docking to the target pocket. **TacoGFN+FT** fine-tunes a pre-trained TacoGFN to tailor the model to the target pocket using docking as the reward.

### 5.4 Experimental results

Table 1 compares TacoGFN against other baselines in the generative problem setting for SBDD. We examine the docking score performances for individual pockets in Figure 4 and 5. We also show examples of generated molecules in Figure 3 and analyze their average physical properties in Table 2. We then compare TacoGFN+FT against existing methods for the optimization problem setting in Table 3 and 4, where docking is used in optimization rounds. Lastly, we conduct ablation studies to show the benefits of using a larger docking score dataset in Table 5 and validate the utility of the pocket conditioning in Table 6.

**Generative setting.** Table 1 highlights the strong performance of TacoGFN in the generative problem setting for the CrossDocked test set pockets. Notably, TacoGFN boasts a significant improvement in success rate at 56.0%, more than doubling the previous best of 24.5% achieved by DecompDiff. This improvement is due to TacoGFN's ability to generate high-affinity molecules that simultaneously satisfy the drug-likeness and ease-of-synthesis requirements. In fact, TacoGFN records the best QED, SA, and High Affinity simultaneously among all **generative** methods. In Figure 3, we show examples of molecules generated by TacoGFN and show its ability to generate molecules with significantly improved docking scores compared to native ligands.

It is difficult to discover molecules that simultaneously exhibit better QED and Vina Dock compared to known binders and molecules from existing baselines due to the trade-off between Vina Dock and QED. Molecules with higher molecular weight are more likely to have strong Vina Dock because of the presence of more interacting atoms, but they tend to have worse drug-like properties (QED). [4] In Figure 4, we show the performance breakdown for the 100 test protein pockets. TacoGFN achieves better average Top-10 Vina Dock than DecompDiff in 57% of the test pockets. Notably, the top molecules generated by TacoGFN

---

[4]This trade-off is clearly observed with Pocket2Mol and DecompDiff (see Table 1 and 2), where DecompDiff has strong Vina Dock but lower QED, while Pocket2Mol exhibits the opposite trend.

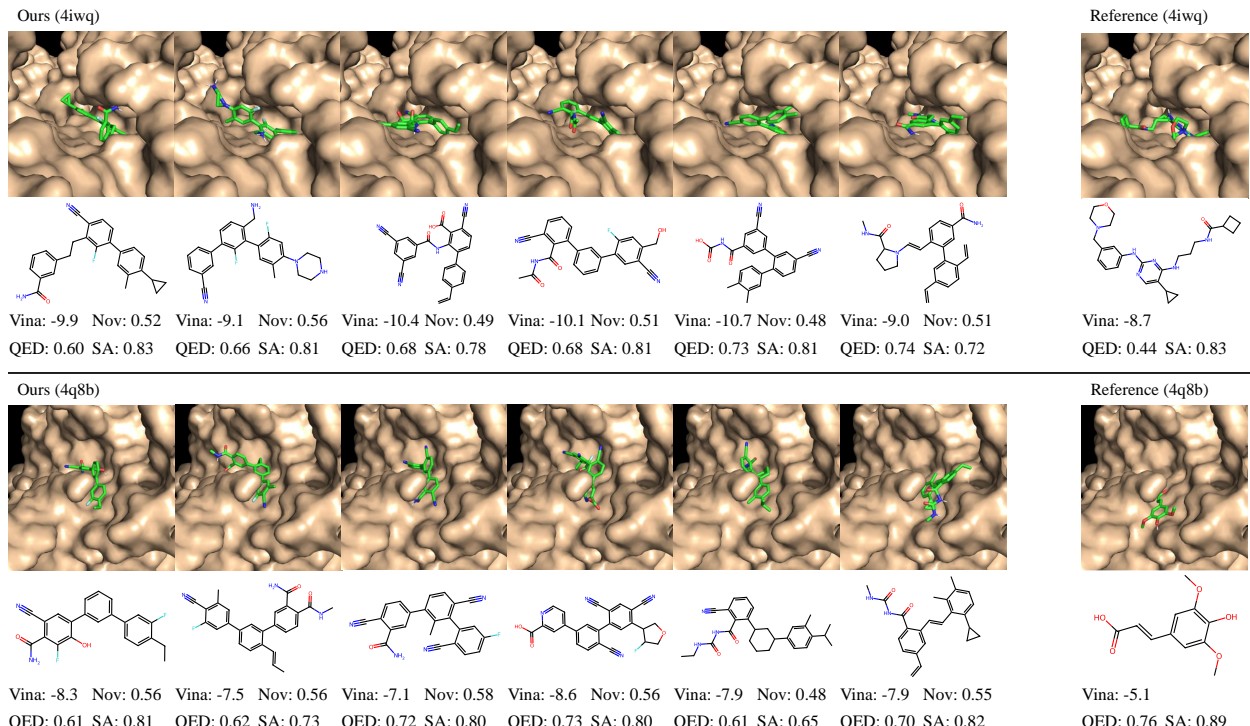

Figure 3: Our method is focused on de novo hit discovery - finding novel and diverse high-scoring hits for a protein target. Our method does not optimize based on a given reference seed compound. The goal of reward-based sampling is to sample diverse high-scoring molecules. To provide a fair overview of the model's performance, we selected protein pockets *4iwq* and *4q8b*, which are at the 25th and 75th percentiles, respectively, based on their docking scores with their native ligands. We show compare the molecules generated by TacoGFN against the native ligand with their QED, SA, Novelty, and Docking score (Vina).

also consistently demonstrate higher QED values. As shown in Appendix Figure 5, in the pockets where DecompDiff achieves a lower Top-10 Vina Dock, the molecules often exhibit a molecular mass larger than 500 daltons. These heavier molecules violate the ideal properties of orally active drugs according to the Rule of 5 (Lipinski et al., 1997). In addition, heavy molecules with high docking scores are more likely to be false positives (Pan et al., 2002). TacoGFN is able to find molecular spaces that improve both QED and Vina Dock requirements not only by modelling the reward distribution but also by exploring a broader chemical space. This is achieved through learning from generated examples using our online policy, rather than being limited to the training data.

We note exploring millions of molecules online could not be easily achieved with existing SBDD baseline. This is because the generation process of TacoGFN is a few orders of magnitude faster than existing autoregressive or diffusion-based generative methods. Additionally, TacoGFN achieves 100% in validity and uniqueness, demonstrating the efficiency of our fragment-based 2D generation framework. As shown in Table 1, TacoGFN achieves a significant improvement in both time and validity over previous methods.

Molecules generated from TacoGFN have more ideal molecular weight and drug-likeness properties, in addition to achieving better Vina Dock; Therefore, they are more suitable as drug candidates. TacoGFN samples at reward temperature $\beta$ of 64 at inference time, resulting in the modelling of the probability $p(x|c) \propto R(x,c)^{64}$. This policy focuses more on the modes of reward function. Therefore the molecules sampled have higher quality but lower diversity than those generated by the methods that do not attempt to satisfy multiple objectives. By changing $\beta$ or reward function we can trade off rewards with diversity as shown in Appendix D.4.2.

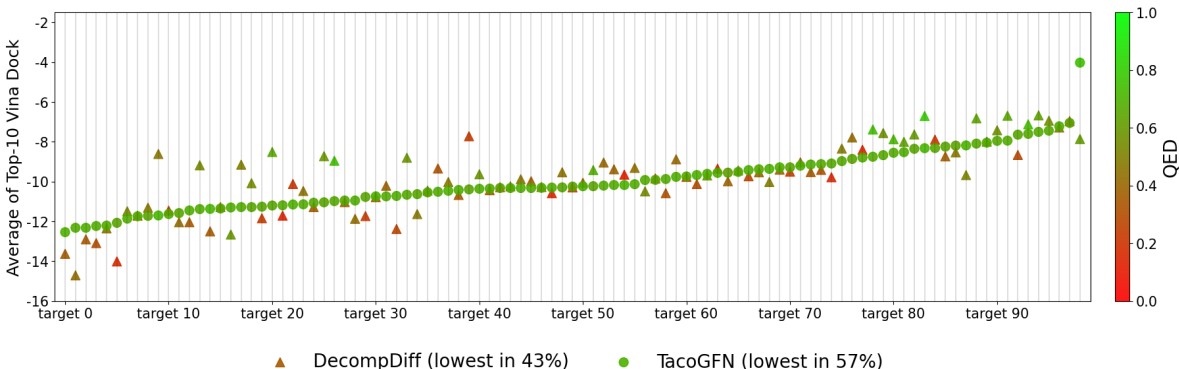

Figure 4: The average of the top-10 Vina Dock of molecules generated for individual CrossDocked test pockets (target) by DecompDiff and TacoGFN. Targets are sorted by the average of the top-10 docking score of TacoGFN generated molecules. A lower docking score means a higher estimated binding affinity. Color is used to denote the average QED value of molecules in the Top-10 set. A higher QED indicates the molecule is more drug-like.

**Evaluation of physical properties.** We evaluate the following physical properties important for small molecule drugs in Table 2: (1) **Ideal Mol. Wt.** is the percentage of molecules satisfying molecular weight within 160 - 480 daltons - the typical acceptable molecular weight range for small molecule drugs (Ghose et al., 1999). (2) **Mol. Wt.** is the molecular weight of molecules in Daltons. (3) **Num Heavy Atom** is the heavy atom (any atom that is not hydrogen) count of the molecules generated. (4) **Strain Energy** is the difference between the re-docked poses (meaning generated molecules are docked with RDKit's ETKDG conformation initialization), versus the strain energy of their unstrained ETKDG poses. We consider the sum of torsional, steric and electrostatic strains. If the strain energy of the ligand's docked pose is very high, the physical compatibility for such binding becomes poor (Perola & Charifson, 2004; Gu et al., 2021).

For baseline, we compare against the top three generative methods ranked by success rate: **DecompDiff** Guan et al. (2023b), **Pocket2Mol** (Peng et al., 2022) and **TargetDiff** Guan et al. (2023a). Additionally, we sample a random set of molecules from the ZINC virtual library for comparison.

Table 2: Comparison on the number of heavy atoms, strain energies

| Model | Ideal Mol. Wt. (↑) Avg. | Mol. Wt. Avg. | Med. | Num Heavy Atom Avg. | Med. | Strain Energy (↓) Avg. | Med. |
|---|---|---|---|---|---|---|---|
| ZINC molecules | 98.73% | 337.63 | 337.34 | 23.53 | 23.50 | 377.77 | 375.17 |
| Pocket2Mol | 64.55% | 248.65 | 230.99 | 18.29 | 16.86 | 254.21 | 218.52 |
| TargetDiff | 78.37% | 323.64 | 327.31 | 22.78 | 23.15 | 600.31 | 547.86 |
| DecompDiff | 51.18% | 494.05 | 487.51 | 34.93 | 34.00 | 834.72 | 781.34 |
| TacoGFN (ours) | 99.76% | 402.56 | 402.65 | 30.47 | 30.52 | 360.51 | 357.33 |

As shown in Table 2, 99.76% of molecules generated by TacoGFN have molecular weights within the 160 to 480 dalton range - the ideal range for small molecule drugs. This ratio is much higher than those of existing methods such as DecompDiff (51.18%), Pocket2Mol (64.55%) and TargetDiff (78.37%). For Pocket2Mol, molecules fall outside of the ideal molecular weight range because they are too small (below 160 Daltons); While for DecompDiff, most molecules are outside of range because they are too heavy (greater than 480 Daltons). We note there are indeed drug candidates with molecular weights or drug-likeness outside of the typical acceptable range. However, they may be regarded as an exception rather than the norm (from a specialized drug class) or have elevated risks for poor absorption and bioavailability (Ritchie & Macdonald, 2014).

Table 3: Comparison of the properties of molecules generated in the **optimization** problem setting for the CrossDocked test set pockets. The prior method results are taken from their publication. The number of fine-tuning steps is 300 unless otherwise specified.

| Model | Validity (↑) Avg. | Vina Dock (↓) Avg. | Med. | High Affinity (↑) Avg. | Med. | QED (↑) Avg. | Med. | SA (↑) Avg. | Med. | Diversity (↑) Avg. | Med. | Success Rate (↑) Avg. | Time (↓) Gen + Score. |
|---|---|---|---|---|---|---|---|---|---|---|---|---|---|
| Reference | 100% | -7.45 | -7.26 | - | - | 0.48 | 0.47 | 0.73 | 0.74 | - | - | 25.0% | - |
| RGA | - | -8.01 | -8.17 | 64.4% | 89.3% | 0.57 | 0.57 | 0.71 | 0.73 | 0.41 | 0.41 | 46.2% | - |
| TargetDiff+Opt | - | -8.30 | -8.15 | 71.5% | 95.9% | **0.66** | **0.68** | 0.68 | 0.67 | 0.31 | 0.30 | 25.8% | >3728 |
| DecompOpt | - | -8.98 | -9.01 | 73.5% | 93.3% | 0.48 | 0.45 | 0.65 | 0.65 | 0.60 | 0.61 | 52.5% | 9241 |
| EvoSBDD ($\alpha = 1.3$, 140R) | 100% | -10.27 | -10.36 | 96.5% | 100% | 0.53 | 0.52 | 0.75 | 0.77 | **0.63** | **0.63** | 78.8% | 6300 |
| EvoSBDD ($\alpha = 0, \sigma = 1$, 140R) | 100% | -10.14 | -10.27 | 94.4% | 100% | 0.59 | 0.59 | 0.77 | 0.77 | 0.62 | 0.62 | 86.4% | 6300 |
| TacoGFN+FT ($t_{QED} = 0.4, t_{SA} = 0.6$) | 100% | **-10.78** | **-10.93** | **97.1%** | 100% | 0.47 | 0.47 | 0.70 | 0.69 | 0.62 | 0.62 | **87.8%** | 7750 |
| TacoGFN+FT ($t_{QED} = 0.5, t_{SA} = 0.8$) | 100% | **-10.33** | **-10.39** | 97.0% | 100% | 0.58 | 0.58 | **0.82** | **0.82** | 0.61 | 0.61 | **88.8%** | 6230 |
| TacoGFN+FT ($t_{QED} = 0.6, t_{SA} = 0.8$) | 100% | -10.14 | -10.21 | 96.2% | 100% | 0.65 | 0.65 | **0.82** | **0.82** | 0.59 | 0.59 | 86.3% | 6140 |

PDBbind (Liu et al., 2014), the parent database of the CrossDocked set (Francoeur et al., 2020), contains a large number of common biomolecules such as ATP (molecular weight: 508 daltons) or sulphate ion (molecular weight: 96 daltons). This inherent bias in the training dataset hinders data distribution learning-based generative models from generating molecules that would be acceptable as real drugs. In contrast, TacoGFN learns the actual physical property distribution of drugs, resulting in an average molecular weight (402.56 daltons) of the generated molecules that closely matches that of FDA-approved drugs ($\sim 430$ daltons).

With regards to strain energy, TacoGFN attains far lower strain energy than DecompDiff and Target-Diff. The strain energy of TacoGFN is comparable with that of ZINC molecules, as shown in Table 2. Pocket2Mol achieves lower strain than ZINC and all other methods, due to the small molecular weight they generate. In summary, this analysis supports that molecules generated by TacoGFN show appropriate physical properties and compatibility with its binding pocket.

**Optimization setting.** Here we study the optimization problem setting, where methods can conduct optimization rounds leveraging docking program on the target pocket. We evaluate TacoGFN+FT, where we fine-tune a pre-trained TacoGFN, using UniDock (Yu et al., 2023) [5] on the target pocket as the affinity reward for generated molecules. For each protein pocket, we chose to finetune TacoGFN for 300 steps (unless otherwise specified) to match the time for TacoGFN+FT ($t_{QED} = 0.5, t_{SA} = 0.8$) with EvoS-BDD's (Reidenbach, 2024). We report the time taken for fine-tuning and docking molecules for our method, measured on a single A4000 GPU - a GPU with less performance than the one used by EvoSBDD (A6000 GPU), or TargetDiff+Opt/DecompOpt (A100 GPU). The details of the fine-tuning process and UniDock can be found in Appendix E.

TacoGFN+FT is able to simultaneously optimize a set of potentially conflicting objectives - consisting of Vina Dock, QED and SA. The TacoGFN+FT with balanced objective ($t_{QED} = 0.5, t_{SA} = 0.8$) simultaneously achieves state-of-art results in Vina Dock, high affinity, synthetic accessibility and success Rate. Since TacoGFN samples molecules during fine-tuning with a lower reward temperature $\beta$ (sampled uniformly between 0 and 64), the diversity of TacoGFN+FT is higher than TacoGFN and comparable with EvoSBDD. Our method variant ($t_{QED} = 0.4, t_{SA} = 0.6$) focusing on affinity attains the best average Vina Dock of -10.78. Our method variant focusing on QED achieves a comparable QED to the best baseline TargetDiff+Opt (0.65 vs 0.66), but a much better Vina Dock (-10.14 vs -8.30) and SA (0.82 vs 0.68). Please see Appendix Table 12 for full results.

Overall, our proposed fine-tuning pipeline is highly efficient, as TacoGFN+FT surpasses state-of-art performance while taking less time. In comparison to EvoSBDD (Reidenbach, 2024), which does not take pocket structure as input, TacoGFN is already trained to generate drug-like and synthesizable molecules with binding conditions to protein pocket structure. Therefore, finetuning TacoGFN generates a molecular set with a better success rate and high-affinity rate for a new protein pocket. Compared to DecompOpt (Zhou et al., 2024), our method does not need to traverse the 500-1000 denoising steps to generate a molecule.

---

[5]Uni-Dock is a recently proposed GPU-accelerated molecular docking program that achieves more than 2000-fold speed-up compared with the AutoDock Vina running in single CPU core.

This means we could search for more molecules and discover higher reward candidates in a shorter amount of time.

Table 4 compares the performance of TacoGFN+FT by simply generating an equal number of molecules with base TacoGFN and ranking the top 100 molecules without finetuning (TacoGFN+Rank); Here we show, although TacoGFN+Rank already attains competitive results, our proposed fine-tuning meaningfully further improves Vina Dock and Success Rate.

Table 4: Comparison of performance between TacoGFN with Finetuning (TacoGFN+FT) with simply ranking the same number of generated molecules from TacoGFN by their reward (TacoGFN+Rank).

| Model | Validity (↑) Avg. | Vina Dock (↓) Avg. | Med. | High Affinity (↑) Avg. | Med. | QED (↑) Avg. | Med. | SA (↑) Avg. | Med. | Diversity (↑) Avg. | Med. | Success Rate (↑) Avg. | Time (↓) Gen + Score. |
|---|---|---|---|---|---|---|---|---|---|---|---|---|---|
| TacoGFN+FT ($t_{QED}=0.5, t_{SA}=0.8$) | 100% | **-10.33** | **-10.39** | **97.0%** | **100%** | 0.58 | 0.58 | **0.82** | **0.82** | **0.61** | **0.61** | **88.8%** | 6160 |
| TacoGFN+Rank | 100% | -10.09 | -10.10 | **97.0%** | **100%** | **0.59** | **0.59** | **0.82** | **0.82** | 0.60 | **0.61** | 86.4% | 5133 |

## 5.5 Ablation studies

We conduct additional ablation studies using the base TacoGFN under the **generative** setting to examine the effect of docking score prediction accuracy and pocket conditioning on our method.

**Effects of using higher quality docking score predictor.** Here, we study the effect of using a more accurate docking score predictor, which is trained on a larger dataset, for reward. Since training a docking score predictor does not require high-quality protein-ligand structural data such as the CrossDock-100k set, we can introduce a second, larger dataset for docking score prediction called **ZINCDock-15M**. It consists of about 15M docking simulation data - from docking 1,000 random ZINC20 (Irwin et al., 2020) molecules into each of the 15,207 unique pockets from CrossDock-100k training split using QVina. We then train our pharmacophore-based docking score predictor on this larger dataset. Please see Appendix Table 11 for a comparison of docking score accuracy between using CrossDock-100k and ZINCDock-15M. As shown in Table 5, TacoGFN using the docking score predictor trained on the larger ZINCDock-15M dataset demonstrates improvements in average Vina Dock, high-affinity rate and success rate. This confirms it is possible to leverage the easily generated large-scale docking score data to generate more novel and higher affinity molecules.

Table 5: Evaluation of *de-novo* drug design performance of TacoGFN using docking score predictors trained on two different docking score datasets.

| Model | Docking score dataset | Vina Dock (↓) Avg. | Med. | High Affinity(↑) Avg. | Med. | Success Rate(↑) Avg. | Med. |
|---|---|---|---|---|---|---|---|
| TacoGFN | CrossDock-100k | -8.24 | -8.44 | 67.5% | 92.0% | 56.0% | 61.5% |
| TacoGFN | ZINCDock-15M | **-8.35** | **-8.53** | **69.5%** | **94.5%** | **58.3%** | **67.5%** |

**Effects of pocket conditioning.** To examine the effect of the proposed pocket conditioning for GFlowNet, we train a molecular generation policy unconditioned on pocket information. The docking score predictor is unchanged - meaning it still predicts a docking score for a molecule with pocket information. The results are shown in Table 6. We observe that the pocket-conditioned GFlowNet achieves higher docking scores compared to the GFlowNet without pocket conditioning. We further measure the number of non-covalent interactions of a molecule to respective pocket, by obtaining the binding pose using QVina (See Appendix D.4.1). We demonstrate generated molecules by pocket-conditioned GFlowNet result in more non-covalent interactions of for all categories (Hydrophobic interactions, Van der Waals contacts, Hydrogen binding) in Table 9. More non-covalent interactions are indicative of the molecule having better specificity to the protein target. This ablation validates that our method is indeed leveraging pocket conditioning to learn a family of molecular distribution across different pocket structures.

Table 6: Effectiveness of pocket structure conditioning evaluated using Vina Dock.

| Model | Docking score dataset | Vina Dock ($\downarrow$) Avg. | Med. | Top-10 Vina Dock ($\downarrow$) Avg. | Med. |
|---|---|---|---|---|---|
| TacoGFN | ZINCDock-15M | **-8.35** | **-8.53** | **-9.97** | **-10.22** |
| w/o pocket conditioning | ZINCDock-15M | -8.04 | -8.18 | -9.65 | -9.81 |

## 6 Conclusion

In this paper, we have investigated the problem of structure-based drug design. To address the limitations of methods based on distribution learning, we have framed pocket-conditioned molecule generation as learning a multi-objective reward distribution using RL. To this end, we propose TacoGFN, a pocket structure conditioned GFlowNet which generate drug-like and high-affinity molecules with respect to any 3D pocket structure. To the best of our knowledge, TacoGFN is the first RL model to address the challenging task of modelling a family of reward functions induced from all pocket structures. Our model effectively explores the greater chemical space, through generating millions of molecules using the online policy during training. We have further introduced a novel pharmacophore-based affinity predictor, where coarse-graining to the protein pocket is shown to achieve more accurate and robust predictions than existing architectures and protein representations. We finally introduce TacoGFN+FT, which fine-tunes the generic TacoGFN for a given test pocket.

Our experiments on the CrossDocked2020 benchmark have demonstrated that TacoGFN and TacoGFN+FT outperform the state-of-the-art methods in terms of Vina Dock, high affinity and success rate. This demonstrates the potential of TacoGFN as a powerful tool for structure-based drug discovery. In future work, we plan to validate the top generated ligands for some clinically relevant protein pockets *in-vitro*, i.e. in wet-lab experiments.

### Broader Impact Statement

This paper presents work whose goal is to advance machine learning methods for drug discovery. Such methods are increasingly being employed in the pharmaceutical industry since they promise to greatly speed-up the lengthy process of drug discovery and to significantly reduce its huge cost. If that promise holds, these machine-learning methods will benefit patients through better care and our society through a reduction of the economic burden of drug development.

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

## A Softwares

In this study, we used the open-sourced code for GFlowNet (Bengio et al., 2021), PharmacoNet (Seo & Kim, 2023) and GVP-GNN (Jing et al., 2021). Our models were implemented using the Pytorch (Paszke et al., 2019) and PyTorch Geometric (Fey & Lenssen, 2019) libraries, which enabled efficient training and evaluation. We utilized RDKit (Landrum et al., 2006), a widely-used chem-informatics library, to handle the molecular structures and compute chemical properties. We employed the QuickVina 2.1 (QVina) (Alhossary et al., 2015) and UniDock (Yu et al., 2023) for docking, and used Openbabel (O'Boyle et al., 2011) and AutoDock Tools (Huey et al., 2012) to generate ready-to-dock files.

## B Problem definition compared to existing RL baselines

TacoGFN addresses the problem of structure-based drug design (SBDD), which aims to have one model generate high-affinity molecules conditioned on any unseen protein structure. We train this one model to generate molecules conditional to different protein target structures in the CrossDocked training set, using predicted affinity, Synthetic Accessibility (SA), and drug-likeness (QED) as the reward. Then we evaluate the trained model's performance on the 100 unseen protein target structures in the CrossDocked test set. Molecules are generated for unseen protein structures during evaluation to measure whether SBDD models have learned generalized protein-ligand interaction patterns during training.

On the other hand, many existing optimization-based RL methods (Bengio et al., 2021; Zhavoronkov et al., 2019; Korshunova et al., 2022; Jeon & Kim, 2020; Goel et al., 2021a; Lee et al., 2023; Reidenbach, 2024) focus on the target-free problem setting - meaning the target protein structure is not used as input conditioning. RGA Fu et al. (2022), DecompOpt Zhou et al. (2024) and TacoGFN+FT use protein structure as input, however, they still require docking oracle calls to the test protein pocket during evaluation to generate a molecule set. Therefore, they also fall within the optimization-based problem setting. These models are trained or optimized to generate molecules which optimize for predicted affinity on one protein target only and then evaluate on the same target. The goal of this evaluation is to measure the effectiveness of the algorithm in terms of generating objects which match a reward distribution. Although both methods use the commonly used metrics such as affinity, Synthetic Accessibility (SA), and drug-likeness (QED) as the reward, the goal of the evaluation and the problem difficulty is different compared to the SBDD setting.

Because protein-ligand interaction is highly specific, the distribution of molecules with high affinity will vary greatly across different possible pocket structures. The SBDD setting, which requires modelling the molecule distribution conditional to any protein pocket and generalizing to unseen protein pockets, is therefore a different and more challenging task than modelling molecule distribution for a single protein pocket.

## C  Method Details

### C.1  Additional details of pocket conditioned GFlowNet

### C.1.1  Molecular generation actions

We follow action used in previous works on fragment-based molecular generation (Jin et al., 2018; Bengio et al., 2021; Hamidizadeh et al., 2023) to construct molecules fragment by fragment. We present the three types of actions available to TacoGFN below:

**(1) FragmentAddition**: At each step, for each fragment node $v_i^L$ in the molecular graph, we apply the same MLP over its node embedding $\mathbf{h}_i^{L(N)}$ which produces logits over the fragment vocabulary. Each logit represents the unnormalized score for attaching fragment node $v_i^L$ to a particular new fragment node $(v_j^L)$ from the vocabulary. These logits correspond to the *FragmentAddition* action type. While the *FragmentAddition* specifies whether fragment node $v_i^L$ connects to fragment node $v_j^L$, it does not specify how they are connected (i.e. which atom on fragment $v_i^L$ forms a bond with which atom on fragment $v_j^L$).

$$\mathbf{a}_{Add} = MLP\left(\mathbf{h}^{L(N)}\right)$$

**(2) AttachmentSpecification** determines how the fragment pairs are connected. At each step, for each directional edge in the molecular graph which connects fragment $v_i^L$ to $v_j^L$, we produce a logit over the atoms of fragment $v_i^L$. Each logit represents the unnormalized score for fragment i connecting to fragment $v_j^L$ via a single bond from a particular atom on fragment $v_i^L$, based on edge embedding $\mathbf{e}_{ij}^{L(N)}$. The molecule can only be completed when all attachment edges are specified.

$$\mathbf{a}_{Attach} = MLP\left(\mathbf{e}^{L(N)}\right)$$

**(3) StopConstruction** is a graph action that marks the finish of a molecule. The logit is produced from a single MLP output based on the final graph embedding $\mathbf{g}^{L(N)}$. All logits are concatenated and scaled into probabilities using the softmax function, and an action is sampled from the distribution. The same process is repeated for each time step until the *stop construction* action is sampled.

$$\mathbf{a}_{Stop} = MLP\left(\mathbf{g}^{L(N)}\right)$$

After all the scores for all possible actions are computed, a final action is sampled based on the equations as follows:

$$P_a(\mathbf{a}|\mathcal{G}_t^L, \mathbf{h}_{\mathcal{G}^{\mathcal{P}}}) = softmax\left(Concat\left(\mathbf{a}_{Add}, \mathbf{a}_{Attach}, \mathbf{a}_{Step}\right)\right)$$
$$a \sim P_a(\mathbf{a}|\mathcal{G}_t^L, \mathbf{h}_{\mathcal{G}^{\mathcal{P}}})$$

### C.1.2 Reward function.

Our reward function consists of properties highly relevant for a drug candidate: Vina Docking Score (DS), Drug Likeliness (QED) and Synthetic Accessibility (SA). Unless otherwise mentioned, $t_{DS} = -8.0$, $t_{QED} = 0.7$ and $t_{SA} = 0.8$ for all models.

**Drug Likeliness (QED) and Synthetic Accessibility (SA):** We first normalize the raw SA score using the formula $\frac{10-SA}{9}$ to obtain a reward between 0 and 1. While QED and SA need to meet a certain threshold to make a good drug molecule, optimizing these values beyond the threshold does not bring additional utility (Coley, 2021). Therefore, we clip reward $r$ for QED or SA to 1 when they achieve their respective threshold $t$. In other words, our model will not be incentivised to optimize QED/SA beyond their threshold value; Therefore more priority will be placed on optimizing the Vina Docking Score when these thresholds are reached.

$$r_{QED} = \min\left(\frac{QED}{t_{QED}}, 1\right)$$

$$r_{SA} = \min\left(\frac{SA}{t_{SA}}, 1\right)$$

**Vina Docking Score (DS):** Here, we define the docking score threshold $t_{DS}$ to -8.0 kcal/mol, corresponding to $1\mu M$ - an important requirement for a drug candidate. Since molecules will not be as useful if they do not surpass this affinity requirement, we scaled down the component of docking score not surpassing threshold $t_{DS}$ by 0.2. This has the effect of reducing rewards for molecules not surpassing this docking score threshold. Lastly, Pan et al. (2002) notes screening based on docking score is biased toward the selection of high molecular weight, as compound size may unfairly contribute to the energy score. We follow their suggestion of normalizing reward by the cube root of heavy atom count (HAC) - to reduce the false positives resulting from the molecular weight bias. Lastly, we multiply the whole term by -1 as our goal is to minimize the Vina Dock score.

$$r_{DS} = -\frac{(DS - t_{DS}) + 0.2 \times \max\left(DS, t_{DS}\right)}{\sqrt[3]{HAC}}$$

We obtain the final reward by multiplying the normalized rewards together:

$$r = r_{DS} \times r_{QED} \times r_{SA}$$

**Model Details.** We use Double GFN (Lau et al., 2023) to improve exploration in sparse reward domains and high-dimensional states. Our model is trained via the gradient descent method Adam (Kingma & Ba, 2017). We list hyperparameters used in Table 7 and compare the training time of our method in Table 10.

## C.2 Additional details of docking score predictor

**Motivation.** When developing a docking score prediction model, two essential requirements are its speed of processing and its applicability to a variety of proteins and ligands. Since the binding poses are computationally or experimentally expensive, previous affinity prediction models or docking score prediction models (Zhang et al., 2023a) predict energy by integrating the 1D representation vectors $\mathbf{z}^{\mathcal{P}}$ of the protein $\mathcal{P}$ and

Table 7: Hyperparameters used for target conditional GFlowNet

| Hyperparameters | Values |
|---|---|
| Num of training steps | $30,000$ |
| Learning rate | $10^{-4}$ |
| Weight decay | $10^{-8}$ |
| Momentum | $0.9$ |
| Adam eps | $10^{-8}$ |
| Sampling $\tau$ | $0.99$ |
| Learning rate $Z-$estimator | $10^{-3}$ |
| Max nodes | $9$ |
| Random action prob | $0.01$ |
| Batch size | $8$ |
| Training reward temp $\beta$ | $Uniform(0,64)$ |
| Inference reward temp | $64$ |
| Pocket cond dim | $128$ |
| Transformer hidden dim | $256$ |
| Num of transformer layers | $2$ |
| QED threshold $t_{QED}$ | $0.7$ |
| SA threshold $t_{SA}$ | $0.8$ |

Table 8: Comparison of training time.

| Model | Total Steps | Batch Size | Total Time (hrs) | Hardware |
|---|---|---|---|---|
| Pocket2Mol | 475k | 8 | 72.6 | GTX A100 80GB |
| TargetDiff | 300k | 4 | 25.0 | GTX A100 80GB |
| DecompDiff | 300k | 4 | 41.7 | GTX A100 80GB |
| TacoGFN | 30k | 8 | 17.7 | RTX 3090 24GB |

$\mathbf{z}^{\mathcal{L}}$ of ligand $\mathcal{L}$, respectively:

$$\mathbf{z}_p = \phi_{\mathcal{P}}(\mathcal{P}) \tag{6}$$

$$\mathbf{z}_l = \phi_{\mathcal{L}}(\mathcal{L}) \tag{7}$$

$$E = \phi_z(Concat(\mathbf{z}_p, \mathbf{z}_l)) \tag{8}$$

However, this approach can not consider the atom pairwise interactions between ligands and proteins due to global pooling, so it shows less generalizability to unseen ligands or proteins. Compared to previous methods, MONN (Li et al., 2020) proposed a pairwise interaction map $\mathbf{I}$ between protein amino acid embeddings $\{\mathbf{h}_i^{\mathcal{P}}\}$ and ligand node embeddings $\{\mathbf{h}_j^{\mathcal{L}}\}$:

$$\mathbf{I} = \{\mathbf{h}_i^{\mathcal{P}}\} \odot \{\mathbf{h}_j^{\mathcal{L}}\} \tag{9}$$

$$E = SumPool(\phi_{\mathbf{I}}(\mathbf{I})) \tag{10}$$

However, the use of full amino acid sequences is computationally expensive for large proteins. Furthermore, in target-based drug design tasks that prioritize binding pocket information, affinity prediction over the entire protein can sometimes lead to incorrect energy calculations for different pockets. Therefore, our study adopts the use of pharmacophores within the binding pocket as an alternative to considering the entire protein sequence. This approach not only effectively captures the unique features of the protein pocket, but also simplifies its topology, thereby addressing the computational burden and the generalization issues. Also, conceptually our docking score predictor determines whether a ligand atom corresponds to each pharmacophore node rather than whether it forms non-covalent interactions with each amino acid or atom.

In the table, we find TacoGFN (C-alpha) outperforms BigBind and DeepBindGCN, indicating that our model architecture (see equation 5 of our paper) which integrates both pairwise interactions and global

pooling information is more effective than the other architectures. Moreover, we also observed that the pharmacophore representation significantly improved the performance over the residue graph representation. Notably, TacoGFN (Pharmacophore) trained with CrossDocked-100k shows better performance on ZINCDock-test compared to the other models trained with ZINCDock-15M. Given that the main challenge of non-structure-based bioactivity predictors is poor generalization performance to ligands not used in the training set Chan et al. (2023), this result demonstrates that the pharmacophore representation of the pocket greatly improves both accuracy and generalization compared to the residue graph representation.

**Training details** For model training, we use 15% of pocket in the training set for the validation set. We used AdamW (Loshchilov & Hutter, 2018) optimizer with betas of (0.9, 0.999) and a weight decay of 0.05, and a learning rate is 0.0001 with a decaying factor of 0.1 per 20,000 iterations. Each iteration has 32 randomly sampled pockets and 128 randomly selected ligands for each pocket during training on ZINCDock-15M. For CrossDock-100k, we used up to 4 ligands per pocket. We optimize our model with SmoothL1Loss (Girshick, 2015) for 100,000 iterations and select the best model weights with the lowest validation loss. The training process takes about a day on 4 NVIDIA RTX A4000 GPUs.

# D    Additional experiment details

## D.1    Dataset

CrossDocked (Francoeur et al., 2020) is a dataset containing 22.5 million poses of ligands docked into multiple similar binding pockets from PDBBind. For **CrossDocked-100k** set, we use an identical processing strategy to previous works (Luo et al., 2021; Peng et al., 2022): First, data points with binding pose RMSD greater than 1Å are filtered. Then, mmseq2 (Steinegger & Söding, 2017) is to cluster data at 30% sequence identity. After splitting, 100,000 protein-ligand pairs are randomly drawn for the training set. 100 test pockets are drawn from the remaining pocket clusters. 15,307 unique protein pockets remain in the training set.

## D.2    Docking protocols

For our protocol under the generative setting, we first convert all generated molecules into SMILES and calculate their ETKDG conformers (srETKDGv3) using RDKit. Then, we prepare ready-to-dock files of ligands and proteins with Openbabel and AutoDockTools (O'Boyle et al., 2011; Huey et al., 2012). Finally, we dock these conformers using Quick VINA 2.1 (QVina) (Trott & Olson, 2010; Alhossary et al., 2015). For QVina, we use a box size of 20 Å and an exhaustiveness of 8. Note for the optimization setting, we adopt UniDock instead following previous works (Reidenbach, 2024) (Please see details in Appendix E.1).

## D.3    Baseline evaluations

For our baseline generation, we selected Pocket2Mol (Peng et al., 2022), TargetDiff (Guan et al., 2023a), DecompDiff (Guan et al., 2023b). We adhered to the default hyperparameter settings for all models. DecompDiff incorporates three prior modes: subpocket, reference, and beta. Of these, we selected the beta mode for our analysis, as it demonstrated the highest docking score, aligning with findings reported in its original publication.

We generated 100 molecules for each of the 100 protein pockets from the Crossdocked2020 (Francoeur et al., 2020) test set. Following generation, we applied a filtering process to ensure that all molecules were unique and met validity criteria. Molecules were deemed invalid and discarded, if they had reconstruction errors, were duplicates, or were disconnected. The remaining SMILES obtained from these models were compiled to establish the baseline. We then proceeded to perform the docking protocol on each of the SMILES to obtain their respective docking scores. Our results were in line with those reported in the original studies of the respective models.

### D.4 Additional Results

#### D.4.1 Additional ablation on effects of pocket conditioning

To further demonstrate the effectiveness of pocket conditioning, we compare the numbers of non-covalent interactions achieved by the generated molecules from pocket-conditioned TacoGFN with TacoGFN without pocket conditioning. The non-covalent interactions between protein and ligand fall into the following categories: (1) **Hydrophobic Interactions** are interactions between nonpolar regions of the protein and the ligand. (2) **Van der Waals Contacts** are weak forces arising from induced electrical interactions between closely positioned atoms of the protein and ligand. (3) **Hydrogen Bonding** refers to strong dipole-dipole interactions between an electronegative atom and (typically) a hydrogen atom. (4) **Total Interactions** refers to the total number of interactions between the protein and the ligand, including all types of non-covalent interactions. The generated molecules are docked using QVina (Trott & Olson, 2010; Alhossary et al., 2015) to generate a binding conformation. We use PoseCheck (Harris et al., 2023) to compute the number of interactions.

In table 9, we found that pocket conditioning increased all types of non-covalent interactions between protein and generated molecules. this result suggests that the pocket structural information can provide guidance in selecting functional groups that can form better supramolecular interactions with the target pocket.

Table 9: Effectiveness of pocket structure conditioning measured using the average numbers of non-covalent interactions (NCI) achieved after docking the generated molecules into the target pocket using QVina. Higher numbers of interactions indicate a stronger binding interaction.

| Model | Docking score dataset | Hydrophobic ($\uparrow$) | Van der Waals contacts ($\uparrow$) | Hydrogen bonding ($\uparrow$) | Total interactions ($\uparrow$) |
|---|---|---|---|---|---|
| TACoGFN | ZINCDock-15M | **5.67** | **7.23** | **1.15** | **14.05** |
| w/o pocket conditioning | ZINCDock-15M | 5.19 | 6.83 | 1.11 | 13.14 |

#### D.4.2 Discussion on diversity

In our experiments with GFlowNet, we find a trade-off between diversity and the goal of optimizing for a higher average affinity of the generated candidate set. As TACoGFN seeks to generate molecules with a strong affinity to a specific pocket structure while optimizing for QED and SA, the resulting solution set is inevitably smaller, leading to reduced diversity.

We note that the molecular diversity of TACoGFN, if desired, can be increased in various ways. Firstly, the reward is currently exponentiated by reward temperture $\beta$. As $\beta$ increases, the reward density is concentrated close to the modes with high reward, and subsequently, there's a reduction in diversity (Jain et al., 2023). Diversity can be improved by lowering the $\beta$.

Table 10: Comparison of average Vina Dock and diversity of TACoGFN trained under different reward settings.

| Model | Vina Dock ($\downarrow$) | Diversity ($\uparrow$) |
|---|---|---|
| TACoGFN | **-8.24** | 0.53 |
| TACoGFN$_{lenient}$ | -7.91 | **0.73** |

In addition, we have experimented with various reward function settings. In the reward scenario adopted in the paper - the strict setting (see Appendix C.1.2), we reduce the reward given to the component docking score that is below -8.0 kcal/mol. We have found that the stricter reward produces better affinity metrics at the expense of lower diversity. In the lenient setting, where we simply reward docking scores based on their normalized raw values, TACoGFN$_{lenient}$ still outperforms Pocket2Mol and TargetDiff in terms of average docking score (-7.91) while having higher diversity (0.73).

In summary, if the key concern is candidate diversity and novelty, TACoGFN can be trained to generate generally high reward molecules under the lenient setting and/or low $\beta$. If the goal is to generate the best

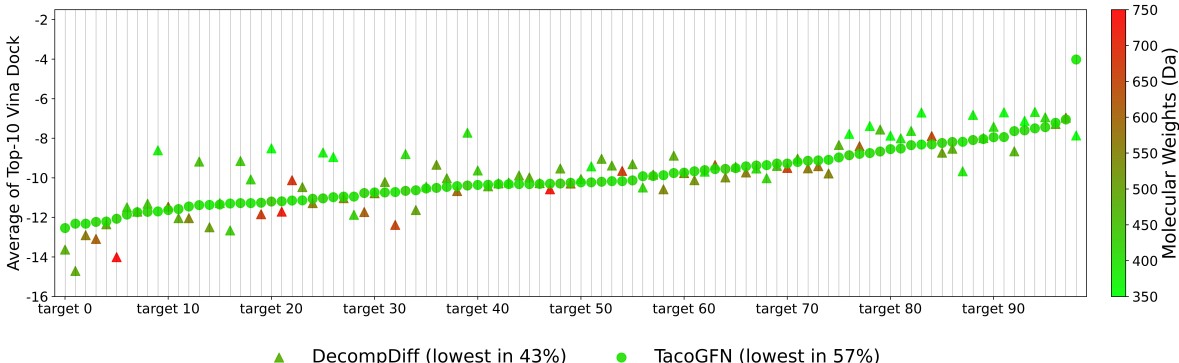

Figure 5: The average of the top-10 Vina Dock of molecules generated for individual CrossDocked test pockets (target) by DecompDiff and TacoGFN. Color is used to denote the average molecular weight of molecules in the Top-10 set. Molecular mass of an orally active drug should be less than 500 daltons Lipinski et al. (1997); Heavy molecules with high docking scores are more likely to be false positives (Pan et al., 2002). Overall TacoGFN consistently achieves ideal molecular weight and strong Vina Dock.

compound set with high average affinity and top affinity with decent diversity, and satisfying QED and SA, TacoGFN can be rewarded under the strict setting and/or high $\beta$.

### D.4.3  Performance of docking score prediction model

While many existing affinity-prediction architectures either work for one target only (Bengio et al., 2021; Gentile et al., 2020), or require binding complex structures (Shen et al., 2022), we cannot adopt these model architectures as we require affinity prediction across various protein targets without using their binding complex structure. Thus, we compare our pharmacophore-based affinity prediction architecture with BigBind (Brocidiacono et al., 2022) and DeepBindGCN (Zhang et al., 2023a), two recently proposed non-complex-based methods which achieved state-of-the-art performance in virtual screening and affinity prediction, respectively. In addition, we examine the effect of introducing pharmacophore-based coarse-grained encoding of the protein pocket, by creating an ablation where we use residue graphs with C-alpha as nodes (as typically done in other methods) Lu et al. (2022) instead of pharmacophore graphs to represent our protein pocket. We call this ablation model TacoGFN (C-alpha). We re-train these models using the same CrossDocked2020 and ZINCDock-15M datasets and evaluate them on both test sets.

Table 11: The evaluation of docking score prediction performance of predictor trained on two different docking score datasets.

| Model | Training set | CrossDocked-100k-test | | ZincDock-15M-test | |
| --- | --- | --- | --- | --- | --- |
| | | RMSE | MAE | RMSE | MAE |
| BigBind | CrossDocked-100k | 1.252 | 0.991 | 1.516 | 1.175 |
| | ZINCDock-15M | 1.561 | 1.227 | 1.233 | 0.955 |
| DeepBindGCN | CrossDocked-100k | 1.739 | 1.366 | 1.912 | 1.531 |
| | ZINCDock-15M | 1.877 | 1.470 | 1.409 | 1.066 |
| TacoGFN (C-alpha) | CrossDocked-100k | 1.091 | 0.821 | 1.517 | 1.157 |
| | ZINCDock-15M | 1.483 | 1.156 | 1.170 | 0.872 |
| TacoGFN (Pharmacophore) | CrossDocked-100k | **0.881** | **0.652** | **1.121** | **0.791** |
| | ZINCDock-15M | **1.143** | **0.880** | **0.862** | **0.574** |

To evaluate the performance of our docking score predictor, we use two test sets: **CrossDocked-100k-test** and **ZINCDock-test**. **CrossDocked-100k-test** is the test set of CrossDocked-100k dataset used in

(Luo et al., 2021) and contains one ligand per each pocket. We re-docked each test ligand to perform the evaluation in the same environment as the generation process. **ZINCDock-test** is the docking simulation data of randomly selected 100 ZINC20 molecules for each pocket in CrossDocked-100k-test. The size of the test sets is 100 pocket-ligand pairs for the CrossDocked-100k-test and 10,000 pocket-ligand pairs for the ZINCDock-test.

# E Finetuning of TacoGFN

Table 12: TacoGFN+FT ablations: Here, we vary the reward function parameters.

| Model | Validity (↑) Avg. | Vina Dock (↓) Avg. | Med. | High Affinity (↑) Avg. | Med. | QED (↑) Avg. | Med. | SA (↑) Avg. | Med. | Diversity (↑) Avg. | Med. | Success Rate (↑) Avg. | Time (↓) Gen + Score. |
|---|---|---|---|---|---|---|---|---|---|---|---|---|---|
| Reference | 100% | -7.45 | -7.26 | - | - | 0.48 | 0.47 | 0.73 | 0.74 | - | - | 25.0% | - |
| RGA | - | -8.01 | -8.17 | 64.4% | 89.3% | 0.57 | 0.57 | 0.71 | 0.73 | 0.41 | 0.41 | 46.2% | - |
| TargetDiff+Opt | - | -8.30 | -8.15 | 71.5% | 95.9% | **0.66** | **0.68** | 0.68 | 0.67 | 0.31 | 0.30 | 25.8% | >3728 |
| DecompOpt | - | -8.98 | -9.01 | 73.5% | 93.3% | 0.48 | 0.45 | 0.65 | 0.65 | 0.60 | 0.61 | 52.5% | 9241 |
| EvoSBDD ($\alpha = 1.3$, 140R) | 100% | -10.27 | -10.36 | 96.5% | **100%** | 0.53 | 0.52 | 0.75 | 0.77 | **0.63** | **0.63** | 78.8% | 6300 |
| EvoSBDD ($\alpha = 0$, $\sigma = 1$, 140R) | 100% | -10.14 | -10.27 | 94.4% | **100%** | 0.59 | 0.59 | 0.77 | 0.77 | 0.62 | 0.62 | 86.4% | 6300 |
| TacoGFN+FT ($t_{QED} = 0.40, t_{SA} = 0.60, n = 240$) | 100% | -10.63 | -10.78 | **97.1%** | **100%** | 0.48 | 0.47 | 0.72 | 0.71 | 0.62 | 0.62 | 86.5% | 6200 |
| TacoGFN+FT ($t_{QED} = 0.50, t_{SA} = 0.80, n = 240$) | 100% | -10.24 | -10.31 | 97.0% | **100%** | 0.58 | 0.58 | 0.82 | 0.82 | 0.61 | 0.61 | 87.7% | 4980 |
| TacoGFN+FT ($t_{QED} = 0.50, t_{SA} = 0.75, n = 240$) | 100% | -10.21 | -10.31 | 96.2% | **100%** | 0.58 | 0.57 | 0.80 | 0.81 | 0.62 | 0.61 | 86.9% | 5210 |
| TacoGFN+FT ($t_{QED} = 0.55, t_{SA} = 0.80, n = 240$) | 100% | -10.16 | -10.27 | 96.8% | **100%** | 0.62 | 0.61 | 0.82 | 0.82 | 0.60 | 0.60 | 86.8% | 5010 |
| TacoGFN+FT ($t_{QED} = 0.60, t_{SA} = 0.80, n = 240$) | 100% | -10.07 | -10.11 | 95.8% | **100%** | 0.65 | 0.64 | 0.82 | 0.82 | 0.60 | 0.60 | 85.7% | 4910 |
| TacoGFN+FT ($t_{QED} = 0.40, t_{SA} = 0.60, n = 300$) | 100% | **-10.78** | **-10.93** | **97.1%** | **100%** | 0.47 | 0.47 | 0.70 | 0.69 | 0.62 | 0.62 | 87.8% | 7750 |
| TacoGFN+FT ($t_{QED} = 0.50, t_{SA} = 0.80, n = 300$) | 100% | -10.32 | -10.39 | 97.0% | **100%** | 0.58 | 0.58 | 0.82 | 0.82 | 0.61 | 0.61 | **88.8%** | 6230 |
| TacoGFN+FT ($t_{QED} = 0.50, t_{SA} = 0.75, n = 300$) | 100% | -10.31 | -10.41 | 96.2% | **100%** | 0.57 | 0.57 | 0.80 | 0.80 | 0.62 | 0.62 | 87.7% | 6520 |
| TacoGFN+FT ($t_{QED} = 0.55, t_{SA} = 0.80, n = 300$) | 100% | -10.24 | -10.35 | 97.0% | **100%** | 0.62 | 0.61 | **0.83** | **0.82** | 0.60 | 0.60 | 87.3% | 6270 |
| TacoGFN+FT ($t_{QED} = 0.60, t_{SA} = 0.80, n = 300$) | 100% | -10.14 | -10.21 | 96.2% | **100%** | 0.65 | 0.65 | 0.82 | 0.82 | 0.59 | 0.59 | 86.3% | 6140 |

## E.1 Finetuning settings.

For finetuning TacoGFN, we adopt the same reward function as described in appendix section C.1.2. However, instead of using the previously mentioned docking score predictor for affinity reward, we use the docking program directly. Following the same setting as EvoSBDD Reidenbach (2024), we adopt UniDock (Yu et al., 2023) - a GPU-accelerated docking program to compute docking score between the generated molecule and the protein target during the fine-tuning stage. We use UniDock's default balanced mode, which has an exhaustiveness of 384 and a max step of 40. We fine-tune using a batch size of 64 to leverage the speedup from UniDock's parallelization for computing docking scores.

For a fair comparison with the previous methods, we re-scored the UniDock's docking pose with Vina (in a score-only mode without any structural changes) and obtained the same values within the margin of error ($\pm$ 0.005 kcal/mol). The top 100 molecules with the highest rewards from the fine-tuning stage are used for evaluation.

## E.2 Results.

For ablations studies, we vary our reward function by adjusting $t_{QED}$ and $t_{SA}$. Once the threshold for $t_{QED}$ or $t_{SA}$ is reached for molecules, the model will not be incentivised to optimize its QED or SA further. With a lower threshold standard ($t_{QED}$ and $t_{SA}$), our model will prioritize optimizing for Vina Dock instead of optimizing for QED or SA. (See details in Appendix C.1.2). We observe such a trade-off between QED and SA with Vina Dock in our ablation studies. We also experiment with various fine-tuning steps $n$. We show that Vina Dock metric will continue to improve with more fine-tuning steps.

