# OpenReview forum: "TacoGFN: Target-conditioned GFlowNet for Structure-based Drug Design"
_TMLR — Accepted by TMLR_

### Review · Reviewer_dEZb · 2024-06-22

**Summary Of Contributions:**

This paper builds on prior work on conditional GFlowNet and multi-objective GFlowNets and applies the framework to the application molecule generation in drug discovery. The authors propose to model the reward distribution conditioned on pocket structure to promote learning across protein targets.

**Audience:**

Yes

**Claims And Evidence:**

Yes

**Requested Changes:**

1. In the conclusion of the paper, the authors claim that "we have framed pocket-conditioned molecule generation as learning a multi-objective reward distribution." However, according to the writing of Section 3, it is unclear why this is a multi-objective problem:
- In Jain et al (2023), the variable w is a d-dimensional vector, however, the parameter c here in the paper seems to represent only one pocket structure: on Page 3 first paragraph under Section 3, the authors write " each conditional information, denoted as c \in C, induces a unique reward function R(x|c)."
- Similarly, for the problem to be a multi-objective problem, c in Equation (1) needs to be a high-dimensional binary vector.
2. The authors often use the word "policy" when they really mean probabilities. Conventionally, a policy refers to a mapping from state to action, so the policy would correspond to the function rather than the value of the function at a point. E.g.,  on page 4 under Equation (1), the authors write "our goal is to learn a policy \pi(x|c,\beta)..."
- additionally, the authors could change the phrase "forward policy" to "forward probability function" (in the first sentence of the paragraph containing Eq(1)) to avoid confusion between the conventional mapping that refers to from state to action.

Section 4 seems to be authors' main contributions. However, it does seem that these methods have all been developed previously from other papers. I am willing to change my decision if the authors can convince me how the approaches taken in Section 4 are novel.

How do the results in Section 5 differ from those in Bengio et al. 2023? It does feel that most of the metrics utilized here were covered by Bengio et al. as well. A detailed contrast between the experimental setups could be beneficial.

For the paper to get published, the authors need to focus on the novelty of the paper and highlight the differences between the proposed experiments and those of the prior work.

**Strengths And Weaknesses:**

The paper provides a thorough case study on drug discovery with the novelty of incorporating pocket structures. Detailed explanations of how pocket structures are generated were provided in the paper. (I am not an expert on the molecules and hence cannot comment on the validity and novelty of the procedures.)

The paper lacks methodological contributions in GFlowNet preliminaries and can benefit from further clarification. Please see the detailed comments on the requested changes.

Overall, the paper feels like reapplying the existing method to a common application. Although the authors named their proposed method TacoGFN, it closely resembles those proposed previously (see citations below). Additionally, GFlowNet and its variants had been extensively applied to molecule discovery previously as evidenced in
Jain et al. (2023) Multi-objective gflownets, ICML.
Bengio et al. (2021) Flow Network based Generative Models for Non-Iterative Diverse Candidate Generation, NeurIPS 2021.
Bengio et al. (2023) GFlowNet Foundations, JMLR.

---

### Review · Reviewer_VTSJ · 2024-06-26

**Summary Of Contributions:**

This paper proposes a novel method named TacoGFN for structure-based drug design. TacoGFN leverages a conditional flow-based network to generate a drug-like molecule fragment by fragment, with probabilities proportional to the reward of each action. This method partially overcomes the limited dataset size for structure-based drug design, and the conditioning on pocket structure still enables the method to be applied to different pocket structures. Experiments show that the proposed method outperforms previous structure-based and (some) optimization-based methods while being more efficient.

**Audience:**

Yes

**Claims And Evidence:**

Yes

**Requested Changes:**

1.	What is the intuition for multiplying instead of summing up the normalized QED, SA, and Vina Dock scores in the design of the reward function? I think this particular design choice needs more explanation for it to make sense.
2.	I would appreciate some discussion on why in Table 3, the QED score of TacoGFN+Finetuning is consistently lower than that of previous methods, while the Vina Dock and SA score both improves.
3.	Minor. There are two seemingly conflicting notation of $\pi$. In the second paragraph of Section 4, $\pi(L|P,\beta)$ means the probability of constructing molecules L given protein structure P and the exponent $\beta$. On the other hand, in Equation (2), $\pi(a|G_t^L, h_{G^P})$ means a policy from which we sample different actions. The authors are encouraged to clarify whether these notations conflict with each other.

**Strengths And Weaknesses:**

Overall speaking, I’m not very familiar with drug design, so I make my comments mostly from the general deep learning perspective.

Strengths:

1.	The paper is clearly written. The background of optimization-based and structure-based drug design methods is comprehensive. The motivation of using a flow-based network is clearly stated.
2.	Many design choices are justified by referring to previous established works or by providing explanations and ablation studies, e.g., the use of geometric vector perceptrons (GVP) in Section 4.1.
3.	The performance of TacoGFN shows a good improvement over previous structure-based methods, especially in the success rate. At the same time, TacoGFN shows a significant reduction in generation time.

Weaknesses:

1.	The evaluation metrics partially overlap with the reward function. The reward function includes three metrics including QED, SA, and Vina Docking Score (DS). In this way, the generation of molecules is inherently biased towards higher QED, SA, and DS, since the probability of each action (e.g., attaching a new fragment) is proportional to the reward function. Therefore, it may be not surprising that the proposed method shows higher QED, SA, and DS scores in the evaluation, compared to other methods. I would like to see some further discussion or clarification from the authors regarding this issue.
2.	About the sampling of molecular generation actions in Section B.1.1. In the final step, the probability of actions is determined by $\pi(a|G_t^L, h_{G^P}) = softmax(Concat(a_{Add}, a_{Attach}, a_{Stop}))$, where the logits $a_{Add}, a_{Attach},a_{Stop}$ are obtained from three different MLPs. My concern is that if the scale of the weights in one MLP (e.g., in the MLP for the “Add” action) is significantly larger than that of the other two MLPs at some timepoint during training, then it is likely that the policy $\pi(a|G_t^L, h_{G^P})$ will always sample one type of action (e.g., “Add”). Such possibility of imbalanced weights may hamper further training.

---

### Review · Reviewer_3Yyf · 2024-06-28

**Summary Of Contributions:**

In this paper, the authors introduce TacoGFN, the first application of GFlowNets to protein pocket conditioned ligand generation. The method follows the GFN framework of learning a policy to generate combinatorial objects with probability proportional to a reward function. The authors consider a composite objective of normalized proxy scores for druglikeness and synthetic accessibility, and a surrogate function trained on docking data.

**Audience:**

Yes

**Broader Impact Concerns:**

Broader impact statement is sufficient.

**Claims And Evidence:**

Yes

**Requested Changes:**

Related to Table 6, the claims related to pocket-conditioning are interesting and it would be informative to see more evidence that the distributions of samples are similarly conditionally dependent on the pocket.

A brief description of Double GFN, the architecture, and training procedure for the policy, in the main text would be helpful.

As a general statement, the gap between modeling the reward distribution of docking score predictions and ranking “suitable drug candidates” is significant. Wherever possible, focusing on the technical contributions of developing GFNs for pocket-conditioned generation (which are clear) is appreciated.

**Strengths And Weaknesses:**

Strengths
As the first application of GFNs to achieve both pocket conditioning and reward-based sampling for ligand generation, the method is interesting and overall is presented clearly.

The experiments are thorough, given the constraints of the proxy and surrogate functions available for evaluation with CrossDock.
Table 6 shows evidence that pocket-conditioning does improve Vina score, and improvements are not simply coming from generation of more inherently “dockable” molecular graphs.




Weaknesses
One point of confusion, the authors state “Since a molecule’s reward (representing its desirability as a real-world drug) should be the same regardless of its predicted 3D conformation, we represent ligands as 2D graphs here.” And then model the reward in terms of docking scores, which are completely sensitive to 3D conformation. Docking score is also only a surrogate for actual binding affinity, which depends on the accessible conformers of the ligand. What do the authors mean by saying that the reward should be invariant to 3D ligand conformation?

I appreciate the direct visualization of molecules in Fig 3, but it makes the evaluation a bit unclear. Is the goal hit expansion (finding high scoring molecules in a lead series related to the ground truth reference), which the method of including the reward exponentiation and reward-based sampling would suggest? Or is it hit diversification, finding high-scoring, high diversity samples?

---

### Decision · Action_Editor_zxwn · 2024-08-26

**Recommendation:** Accept as is

**Comment:**

The paper was reviewed by three reviewers. After rebuttal, all three reviewers were positive (2 accept and 1 leaning accept).

Initial concerns included lack of clarity in presentation about reward modeling and evaluation, lack of background description on Gflownet in the main text, partial overlap between evaluation metric and reward function, imbalanced weights of the trained networks, and lack of novelty. These concerns were addressed successfully by the rebuttal.

In summary, the paper makes a useful contribution to structure-based drug design.

**Audience:**

The paper can be of interest to researchers working on generative models and AI for drug design.

**Claims And Evidence:**

This paper proposes TacoGFN, a novel GFlowNet-based approach for structure-based drug design, for generating molecules conditioned on protein pocket structure with probabilities proportional to its affinity and property rewards. On CrossDocked2020 benchmark, TacoGFN attains state-of-the-art success rate and median Vina Dock score. Fine-tuning TacoGFN further improves the results, outperforming optimization-based methods.

The claims made in the submission are supported by experimental evidence.